# Symptomatic, Genetic, and Mechanistic Overlaps between Autism and Alzheimer’s Disease

**DOI:** 10.3390/biom11111635

**Published:** 2021-11-04

**Authors:** Muhammad Shahid Nadeem, Salman Hosawi, Sultan Alshehri, Mohammed M. Ghoneim, Syed Sarim Imam, Bibi Nazia Murtaza, Imran Kazmi

**Affiliations:** 1Department of Biochemistry, Faculty of Science, King Abdulaziz University, Jeddah 21589, Saudi Arabia; mhalim@kau.edu.sa (M.S.N.); shosawi@kau.edu.sa (S.H.); 2Department of Pharmaceutics, College of Pharmacy, King Saud University, Riyadh 11451, Saudi Arabia; salshehri1@ksu.edu.sa (S.A.); simam@ksu.edu.sa (S.S.I.); 3Department of Pharmacy Practice, College of Pharmacy, AlMaarefa University, Ad Diriyah 13713, Saudi Arabia; mghoneim@mcst.edu.sa; 4Department of Zoology, Abbottabad University of Science and Technology (AUST), Abbottabad 22310, Pakistan; nazia.murtaza@gmail.com

**Keywords:** autism spectrum disorder, Alzheimer’s disease, symptoms, genes, pathogenesis, mechanisms

## Abstract

Autism spectrum disorder (ASD) and Alzheimer’s disease (AD) are neurodevelopmental and neurodegenerative disorders affecting two opposite ends of life span, i.e., childhood and old age. Both disorders pose a cumulative threat to human health, with the rate of incidences increasing considerably worldwide. In the context of recent developments, we aimed to review correlated symptoms and genetics, and overlapping aspects in the mechanisms of the pathogenesis of ASD and AD. Dementia, insomnia, and weak neuromuscular interaction, as well as communicative and cognitive impairments, are shared symptoms. A number of genes and proteins linked with both disorders have been tabulated, including MECP2, ADNP, SCN2A, NLGN, SHANK, PTEN, RELN, and FMR1. Theories about the role of neuron development, processing, connectivity, and levels of neurotransmitters in both disorders have been discussed. Based on the recent literature, the roles of FMRP (Fragile X mental retardation protein), hnRNPC (heterogeneous ribonucleoprotein-C), IRP (Iron regulatory proteins), miRNAs (MicroRNAs), and α-, β0, and γ-secretases in the posttranscriptional regulation of cellular synthesis and processing of APP (amyloid-β precursor protein) have been elaborated to describe the parallel and overlapping routes and mechanisms of ASD and AD pathogenesis. However, the interactive role of genetic and environmental factors, oxidative and metal ion stress, mutations in the associated genes, and alterations in the related cellular pathways in the development of ASD and AD needs further investigation.

## 1. Introduction

Autism spectrum disorder (ASD) and Alzheimer’s disease (AD) are two neuropathological or neuropsychiatric conditions affecting young and old individuals, respectively [1,2]. Represented by hearing/speech impairments, weak immunity, poor cognitive functioning, weak neuromuscular interaction, insomnia, problems in gastrointestinal functioning, and social interaction, ASD is mostly diagnosed in children between the ages of 18 months and 3 years. Currently affecting around 1 in 58 young children, it is one of the most rapidly growing neurological conditions worldwide [3,4]. Both the pathophysiology and etiology of ASD are poorly understood. However, many environmental, genetic, and mother-and-child associated risk factors for ASD have been reported [5,6]. Alzheimer’s disease (AD) is characterized by dementia, cognitive impairment, and memory loss in elderly populations. According to estimates, 5.8 million US citizens aged 65 or above are battling the disease, with this number expected to exceed 88 million in 2050 [7,8]. Accumulating amyloid-β peptide (Aβ), resulting in plaques and microtubular protein tau-causing neurofibrillary tangles, are responsible for AD [9]. Gradual disintegration of synaptic clefts, atrophy of neurons, and dementia have been linked with the aggregation of Aβ and tau. Disease can be categorized on the basis of these proteins [10,11,12]. Recent investigations have highlighted a few coinciding aspects of ASD and AD, two neurological and psychiatric diseases affecting two entirely different age groups. Both ASD and AD have been characterized by dementia, poor cognitive functions, speech impairments, intellectual disability (ID), and depression [13,14,15,16]. The prevalence and severity of both diseases involve complex interaction between the genetic, epigenetic, and environmental factors. Mitochondrial dysfunction and inflammation have also been reported to play an important role in the progression of ASD [17,18]. Studies conducted on 12 genes of Notch and 31 genes of WNT (wingless) signaling pathways associated with AD have shown common pathophysiology of ASD and AD [19]. Activity-dependent neuroprotective protein (ADNP), associated with autophagy, represents another common target for the management of autism, Alzheimer’s disease, and schizophrenia [20]. Recently, the processing mechanism of amyloid-β precursor protein (APP) along the anabolic and catabolic pathways has described the overlapping pathophysiology of ASD and AD [21]. Recently, many correlated medicinal interventions against the common symptoms in both diseases have been adopted. For example, for aggressive and agitated ASD and AD patients, antipsychotics, including quetiapine and risperidone, have been found effective [22,23,24]. For depression and anxiety, SSRIs (selective serotonin reuptake inhibitors), such as venlafaxine and citalopram, have been recommended [24,25,26]. Furthermore, to manage sleep problems and improve neuron health and physiology, melatonin has been found effective [27,28]. On the basis of recent developments, we intend to present a comprehensive account of mechanistic and interventive overlaps between ASD and AD. 

## 2. Methodology

The present review study was conducted at the digital library of King Abdulaziz University, Jeddah Saudi Arabia. There was no need for ethical approval or permission as no animal or human subjects were directly included. Associated literature about the overlapping symptoms, mechanisms, and medical interventions in autism and Alzheimer’s disease was collected. Based on the information available in the recent literature, the commonalities in the pathophysiology of both diseases were described.

Data were gathered from web databases, including Yahoo, Google Scholar, PubMed, and Web of Science. Terms such as autism, Alzheimer’s disease, genes in autism, genes in Alzheimer’s disease antidepressants, antipsychotics, sleep promoters, autism pathogenesis, and Alzheimer’s disease pathogenesis were used to collect data. Recent data were acquired from peer-reviewed research publications published in reputable journals. Data from websites, unpublished articles, and published articles in non-peer reviewed journals were excluded. Data included in the final document were combined, analyzed, and evaluated. Overall, 278 information sources (published articles, books, and websites) were combined, and 230 were included in the present study.

Prisma flow diagram for systematic review, Figure 1, representing the records identified from all databases searched. The records excluded and included in the present review are described.

## 3. Shared Symptoms of ASD and AD

Both ASD and AD are categorized as neurodevelopmental and neurodegenerative diseases. Several peripheral neuropathies or polyneuropathies that may be the collective effect of mitochondrial disorders have been associated with ASD [29,30,31]. Recently, neurofibromatosis type 1 (NF1) has been associated with the severity of ASD and ADHD (attention-deficit/hyperactivity disorder) [32]. Neuropathologically, AD is caused by the extracellular build-up of amyloid-β (Aβ) plaques and the intracellular accumulation of tau protein, the component of neurofibrillary tangles (NFTs) [33,34]. 

Dementia is a complex term encompassing memory loss, difficulty or inability in reasoning, and diminished problem-solving abilities. Dementia and ASD are closely linked by their symptoms. Many symptoms of dementia, including impairments in social interaction [35], highly complex routines, food selections, and exaggerated response to sensory or physical irritations, are similar to those of ASD [36,37]. Many experts have shown symptomatic identity in dementia and ASD [38]. Recently, in a meta-analysis of 12 studies, researchers declared Alzheimer’s disease the most common type of dementia, although they could not significantly differentiate between AD and dementia [9,39]. Some other studies have considered AD to be one of the major causes of dementia [40]. Hence, dementia is one of the representative overlapping features of AD and ASD. However, further studies are required to delineate between ASD, AD, and dementia.

Most children with autism have very limited language and interaction skills. By contrast, some have very good vocabulary and can discuss a particular topic in detail. However, about 22% of these children lose their vocabulary as they develop [41]. Some children with autism start speaking between the age of 5 and 7, and a few fail to learn a language by the age of 13 years. Early delay in speech and language is more prevalent in male children [42,43]. Alzheimer’s disease is initiated by memory loss, followed by the inability to use language in communication and social interaction. Gradual loss of vocabulary by neurodegeneration is one of the major symptoms of AD that is linked with the loss of cognitive functions [44]. Individuals suffering from AD face more difficulty in denomination tests; they use empty long pauses during discussion. However, the severity of this symptom depends on the stage of disease and is positively correlated with AD progression [45,46,47]. The distribution of speech pauses has been described as a marker for AD diagnosis [48]. Studies have shown that identification of language and diarization of speakers provide promising results for the diagnosis of speech loss in the case of AD [49].

The neurological skills or abilities a person uses to read, remember, concentrate, or make decisions are known as cognitive skills. Having poor cognitive skills is one of the characteristics of children with autism. Some children develop variable cognitive skills with age, but most of them fail to develop a social network [50,51]. The fact that a few develop better cognitive skills compared to other children with autism may be due to compensation mechanisms. According to recent estimates, up to 70% of individuals with ASD have ID (intellectual disability), while only half of those with ID have ASD [52]. Children with symptoms of attention-deficit/hyperactivity disorder (ADHD) along with ASD have shown severe cognitive problems [53]. Generally, a decline in cognitive skills is considered to be a phenomenon of aging, with an increased rate detected after the age of 65 [54,55]. However, this decline in cognitive abilities cannot severely affect normal life activities [56]. Alzheimer’s disease rapidly deteriorates cognitive functions [57]. Plasma D-glutamate levels have been suggested as a marker for the detection of mild cognitive impairments and AD [58].

Anxiety and depression are two major signs of ASD, with up to 84% children with autism having mild to severe anxiety [59,60]. Among autistic children, multiple impairments can be a cause of anxiety. Determination of anxiety levels in children with autism is a difficult process [61,62,63], as the levels increase with age [64,65]. High intensity of anxiety has been reported in children with less severe autism and higher IQ (intelligence quotient) levels [66]. Many strategies adopted by parents to hide autistic traits during social interactions may also increase the level of depression among children with ASD [67]. Alzheimer’s disease is also accompanied by neuropsychiatric symptoms, including depression [68]. According to the meta-analysis of recent studies, up to 16% of AD patients exhibited major depression [69]. AD increases the severity of depression, which in turn induces cognitive impairments [70].

Children with autism experience up to an 80% frequency of sleep disturbance, mainly due to a deficiency of melatonin [71,72]. Restlessness is common among children with ASD [73]. Disturbance in circadian rhythms, sleep apnea, and fragmented nocturnal sleep are also among the core symptoms of AD [74]. Circadian rhythm and sleep duration contribute to the behavioral functions and brain development [75,76]. Interventions for sleep disturbances have been suggested to modulate the symptoms of cognitive impairment [77,78]. According to the recent international criteria for the diagnosis of AD, the disease does not necessarily interfere with any occupational or social functioning at early stages. However, gradual impairments in memory and cognitive functions are prominent signs of AD [79]. Similar diagnostic and staging procedures and biomarkers have been used for AD and early memory impairment [80]. Structural disintegration and loss of neuronal connections have recently been identified by magnetic resonance imaging (MRI). Studies have also shown that physical activity may ameliorate the decline in the functioning of white matter and, therefore, AD progression [81]. Some of the correlated symptoms of ASD and AD are illustrated in Figure 2.

## 4. Genes and Genetics

Numerous genes have been linked with the neurological abnormalities such as ASD and AD. According to some studies, 1007 autism risk genes have been reported. Using the SFARI gene database, 212 genes—environment interacting pairs—have been identified [82,83]. Some examples of the genes associated with autism include MECP2, CHD8, KMT2A, GRIN2B, SCN2A, NLGN1, NLGN3, MET, CNTNAP2, FOXP2, TSHZ3, SHANK3, PTEN, DYRK1A, RELN, FOXP1, SYNGAP1, NRXN [84,85], the NLGN gene series (NLGN1, NLGN2, NLGN4, and NLGN4Y) [86,87], brain function protein-related genes (such as GABRA5, GABRB3, UBE3A, HERC2 and CYFIP1 [88]), NRXN2, POGZ, RFX3, ANK2, ARHGEF10, BRD3, CEP152, CHRM3 [89], STX1A, NLGN3, SHANK2, DLGAP1, NRXN3, DLG4, CACNG2, AKAP9, CACNA1C, KCNS3, CACNA2D3 [90], and some metabolic genes, (such as GSTT1, GSTM1, and GSTP1 [91]). Many of these genes are pleiotropic in nature [92]. Mostly, these genes regulate many other genes. For example, up to 3% of individuals have fragile X syndrome (FXS) caused by mutations in the FMR1 gene—a gene that regulates about 6000 mRNAs in the brain and maintains synaptic plasticity [93]. A set of 805 DNA methylation regions (DMRs) of paternal sperm were demonstrated to predict paternal susceptibility to autism in their offspring [94]. Rett syndrome is another disorder associated with ASD. It mostly occurs in females and is caused by mutations in the MECP2 gene, which regulates several genes in neurons [95]. Williams syndrome and autism have shown a similar pattern of gene expression but different socio-cognitive profiles [96].

Variants of at least three dominant autosomal genes have been reported in Alzheimer’s disease. These genes, which are found responsible for the early onset of AD, include presenilin 1 (PSEN1), presenilin 2 (PSEN2), and amyloid precursor protein (APP), and variations in these genes are clearly associated with the early onset of AD [97,98]. As these genes are autosomal dominant, the disease is generationally inherited. Almost 50% of the children of affected patients have the chance to adopt the mutated allele. Mutations in these three genes result in the enhanced production of AB (amyloid beta) fragments and plaque formation [99]. Polymorphic variations in the gene coding for apolipoprotein E (APOE) results in the formation of three isoforms of protein. However, these variants are not found to be critically involved in the development of AD, as they have 8% to 77% frequency in normal human populations worldwide [100,101]. Some genetic factors associated with ASD and AD are tabulated (Table 1).

## 5. Theories and Mechanism of Pathophysiology

Autism is a neurodevelopmental disease affecting very young children, while Alzheimer’s disease is a neurodegenerative disease affecting elderly people. AD is associated with the shrinking of the amygdala, whereas ASD is represented by an enlarged amygdala region. The association between neurodevelopmental and neurodegenerative diseases is a topic of growing interest. In this section, we discuss some theories and mechanisms to emphasize the correlation or overlaps between these apparently divergent conditions.

### 5.1. Disrupted Neural Connectivity

In the neurotypical brain, old, unwanted, and non-functional neurons are removed to improve the networking and operational efficacy of neural circuits. This ability is missing in ASD children and results in poor interaction between neurons and also between the different regions of the brain. In the case of autism, there is an extensive increase in the number of neurons, resulting in the poor shape and fine-tuning of neurological circuits [181,182,183,184,185]. In the autistic brain, there are flaws in the process of removing poorly functional synapses, resulting in overall dented synaptogenesis with abnormal dendritic spines that hinder the neural coordination. Rett syndrome, which presents this condition, is associated with ASD [186,187]. Rett syndrome accounts for less than 1% of ASD, occurring almost exclusively in girls, while autism is diagnosed four times more frequently in boys. The mTORC1 (mechanistic target of rapamycin (mTOR) complex 1) pathway is differentially regulated in Rett syndrome and other syndromes associated with ASD [188]. In the case of AD, the abnormal accumulation of soluble Aβ results in the impairment of neural circuits and increases the number of hyperactive cells [189]. According to relevant studies, the normal level of soluble Aβ supports synaptic plasticity, while increased levels trigger a toxic cascade that results in synaptic impairment and cognitive deficits [190]. Hence, in both cases, neural connectivity and synaptic structure and physiology are responsible for disturbed neural and brain functions.

### 5.2. Imbalanced Neurotransmitters

Elevated serotonin (5-hydroxytryptamine, 5-HT) levels (hyperserotonemia) have been found in more than 30% of children with autism as compared to normal controls. About 99% of blood serotonin is found in the platelets and 1% in the plasma exposed to catalytic enzymes [191]. Serotonin plays an important role in brain development in younger populations and in maintaining proper physiology in adults, as well as in regulating autonomic, behavioral, and cognitive functions [192]. During pregnancy, serotonin is produced by the placenta from tryptophan obtained from the mother, with the brain beginning its synthesis later on [193]. Variations in the genes that regulate the mechanism of serotonin production lead to abnormal levels in the blood as well as neurodevelopmental abnormalities in the CNS (central nervous system) of patients. On the other hand, reduced levels of serotonin have been found in many areas of the brain in AD patients. Serotoninergic system problems have been implicated in anxiety, depression, aggression, and restlessness in AD patients [194].

Dopamine is one of the most extensively studied neurotransmitters, with an established role in neurological and psychological disorders [195]. According to recent findings, brain dysfunction in autism is associated with overactivated dopamine systems. First, there is the link between hyperdopaminergic activity and autism; second, there is the link between right-hemispheric dysfunction and deficiencies in autism; and third, dopamine activation is associated with an increase in the prenatal and perinatal risk factors that increase the incidence of autism-associated behavioral traits [5]. Emotion processing control in the amygdala is inhibited by the activated dopamine system [196,197]. A meta-analysis of seventeen studies and 512 patients has shown a decreased level of dopamine in AD patients as compared to normal controls, indicating a positive association between decreased dopamine levels and the pathophysiology of AD [198]. Despite being a major neurotransmitter, dopamine has recently been named as a factor in the onset of AD [199]. Glutamate is one of the main neurotransmitters, and iGluRs (ligand-gated ionotropic glutamate receptors) play a crucial role in synaptic plasticity and memory. In the cerebral cortex, glutamatergic neurons comprise up to 80% of the brain’s total metabolic activity under non-stimulated conditions [200,201]. Disruption of iGluRs has been associated with neuropathological conditions, such as Alzheimer’s disease and brain damage. It has also been demonstrated that iGluRs and their regulatory proteins are also altered in ASD and fragile X syndrome [202,203].

### 5.3. Overlapping Mechanisms of Pathogenesis

Autism and Alzheimer’s disease (AD) represent two specific yet correlated disorders. Each of them appears at opposite ends of the human lifespan (early and old age; ASD is represented by an enlarged amygdala, while AD is linked with reduced amygdala); ASD is neurotrophic and AD is neurodegenerative in nature [21,204,205]. Many proteins have been reported to contribute to the pathophysiology of ASD and AD [206]. Mutated or non-functional brain protein SHANK3 is responsible for communication problems in children with autism [207,208]. Downregulation of SHANK3 has also been linked with AD [209]. Fragile X mental retardation protein (FMRP) [164,210,211] and heterogeneous ribonucleoprotein-C (hnRNPC) have also been linked with both diseases [212]. Some RNA binding proteins, including IRP-1 and IL-1, also contribute to the pathogenesis of these diseases [21,213]. However, beta-amyloid precursor protein (APP) plays a central role in the pathophysiology of both ASD and AD [204,214]. According to recent information, the pathogenesis of ASD and AD proceeds in parallel in the context of the synthesis and processing of β-amyloid precursor protein (APP). In general, the synthesis of APP is regulated by several factors. At post-transcriptional levels, many types of miRNAs downregulate protein synthesis. Alterations in these miRNAs have been found in AD, and such miRNAs are a topic of considerable importance in management of the disease [215,216,217]. Iron regulatory proteins (IRPs) bind to the 5′-UTR of mRNA and can regulate the translation in an iron-dependent manner, i.e., by promoting protein synthesis in cells with sufficient iron contents and vice versa [218,219]. FMRP is an RNA-binding protein abundantly produced in brain, and its reduction leads to the onset of fragile X syndrome. The protein binds the APP mRNA in the guanine-rich coding region and represses translation and APP synthesis [220]. FMRP also regulates the export of APP mRNA from the nucleus [216,221]. Another RNA binding protein, heterogeneous nuclear ribonucleoprotein C (hnRNPC), competes with FMRP for the mRNA binding site and upregulates APP synthesis [220,222]. Hence, hnRNPC and FMRP regulate protein synthesis in opposite directions. Therefore, a crosstalk between hnRNPC and FMRP should be present for appropriate homeostasis of APP in the different regions of the brain. FMRP also regulates the cytoplasmic level of α-secretase and γ-secretase and decides the fate of APP processing towards catabolic or anabolic pathways, resulting in AD or ASD, respectively [21,223,224]. The digestion of APP with α-secretase and γ- secretase initiates a non-amyloidogenic pathway, resulting in the production of p3 and s-APPα protein moieties. sAPPα (soluble amyloid precursor protein) persuades an upregulation of glutamatergic synapses and a decline in GABAergic synapses, which results in an excitatory/inhibitory imbalance as found in the case of ASD. Additionally, sAPPα overturns the GABAergic synapses by reducing the release of presynaptic vesicles by direct binding of the sAPP extension domain to GABA type B [225,226]. FMRP and hnRNPC regulate the imbalance between total secreted APP (s-APPα), and Aβ indicates a disturbed iron balance, disrupted neuron density, and inter-neuron transmission, resulting in autism [227,228]. Although the elevated level of s-AAPα is associated with autism, an overall increase in the synthesis of APP is not found in autism [229]. Overgrowth of both white and gray matter in the brain, resulting in an enlarged brain size at an early age in children with autism, is based on neurotrophic activity caused by s-APPα and associated factors [230]. Based on the information available, a parallel yet divergent mechanism of ASD and AD pathogenesis is proposed (Figure 3).

## 6. Conclusions

ASD and AD are neurodevelopmental and neurodegenerative disorders affecting two opposite ends of the human life span, i.e., young children and old people, respectively. Many common symptoms, including speech and language problems, cognitive issues, dementia, reasoning, and planning disabilities, have been observed in both disorders. The common genetic links and genes involved in the pathogenesis of both diseases have been tabulated. However, the pathways involved in the pathophysiology of ASD and AD and the effects of specific gene mutations on the development of these diseases need further investigation. The common factors and physiological conditions have been discussed in detail, and overlapping mechanisms of pathogenesis for both diseases have been proposed. The interaction of genetic and environmental factors, including iron imbalance, oxidative stress, inflammatory cytokines, and their qualitative and quantitative role in the pathophysiology of both disorders also need further study.

## Figures and Tables

**Figure 1 biomolecules-11-01635-f001:**
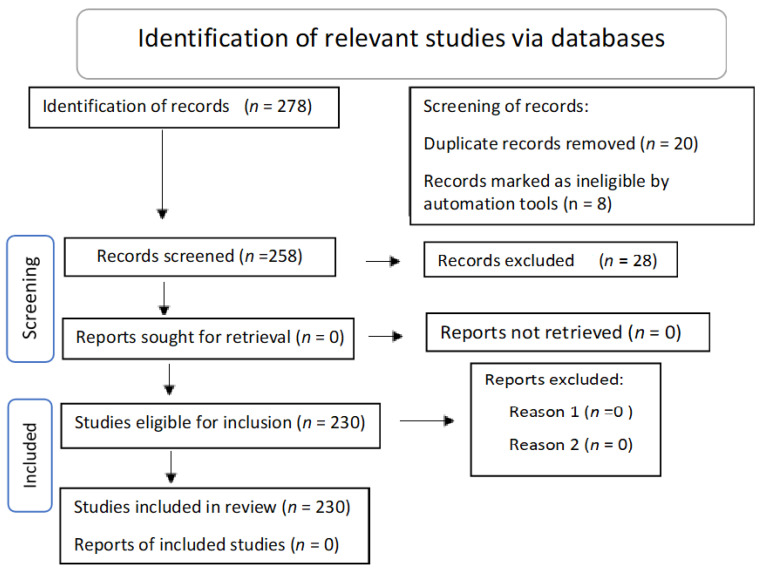
Prisma flow diagram for systematic review.

**Figure 2 biomolecules-11-01635-f002:**
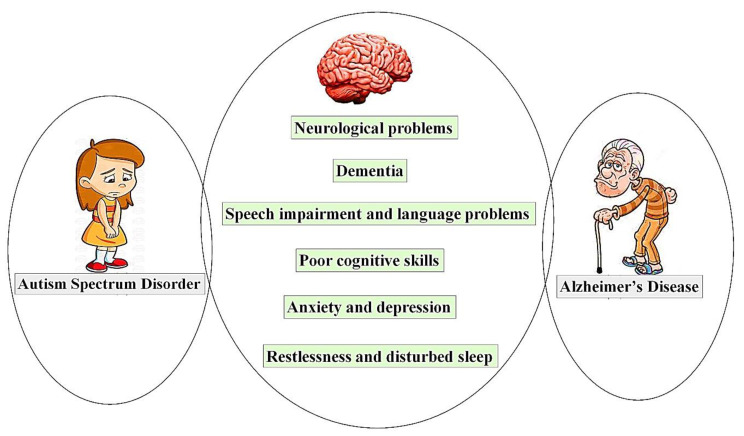
A brief illustration of correlated signs and symptoms representing autism spectrum disorder and Alzheimer’s disease.

**Figure 3 biomolecules-11-01635-f003:**
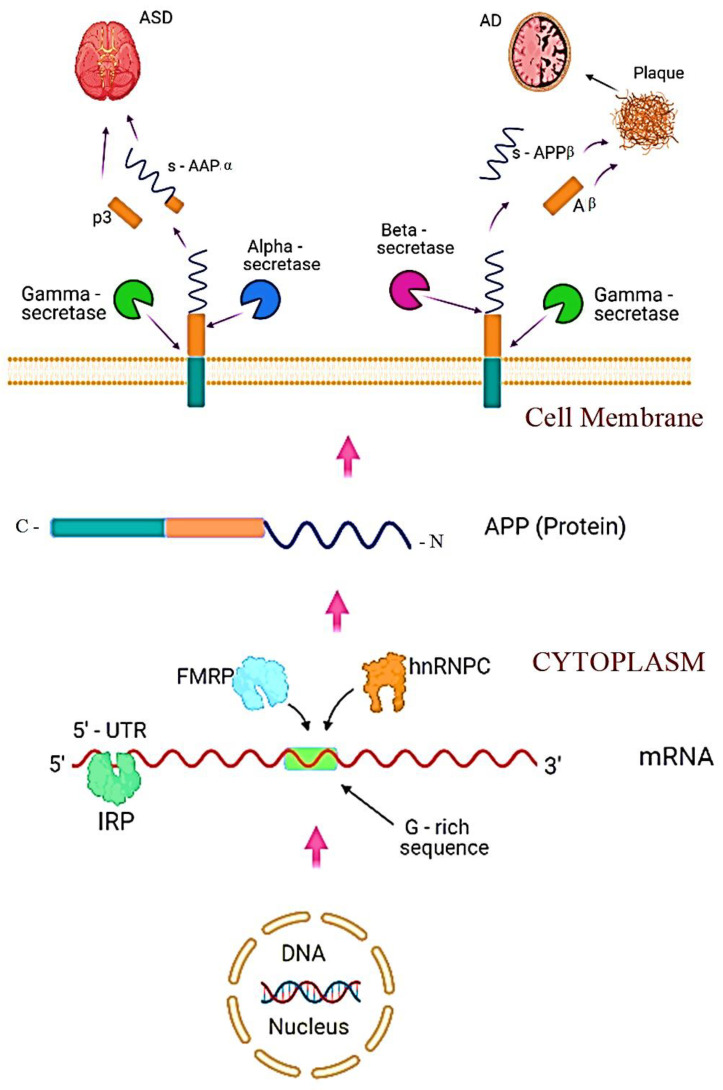
Parallel yet divergent proposed mechanism of ASD and AD pathogenesis. Transport and translation of APP mRNA are well regulated by mRNA transport proteins and RNA binding proteins, such as IRPs and FMRP and miRNAs. There is a G-rich domain in the coding region of mRNA. FMRP and hnRNPC compete to bind this domain to suppress or enhance translation, respectively. APP is a transmembrane protein that is processed by combinations of α-, β-, and γ-secretases, resulting in different product combinations. p3 and s-AAPα promote ASD downstream, while s-APPβ and Aβ, in combination with other molecules, result in the formation of senile plaques and cause AD.

**Table 1 biomolecules-11-01635-t001:** Shared genes and factors associated with ASD and AD.

Sr. No.	Genes Associated with ASD and AD	Reference
1	MECP2 (methyl-CpG binding protein 2)	[102,103].
2	ADNP (activity-dependent neuroprotective protein)	[20,104].
3	GRIN2B (glutamate ionotropic receptor NMDA type subunit 2B)	[105,106,107].
4	SCN2A (sodium voltage-gated channel alpha subunit 2)	[108,109].
5	NLGN (neuroligin)	[110,111,112].
6	CNTNAP2 (contactin-associated protein 2)	[113,114,115,116].
7	TSHZ3 (teashirt zinc finger homeobox 3)	[117,118,119].
8	SHANK	[120,121,122,123,124].
9	PTEN (phosphatase and tensin homolog)	[125,126,127,128].
10	DYRK1A (dual-specificity tyrosine phosphorylation-regulated kinase 1A)	[129,130,131,132].
11	RELN (reelin)	[133,134,135].
12	FOXP1 (forkhead box protein P1)	[136,137,138,139].
13	SYNGAP1 (synaptic Ras GTPase-activating protein 1)	[140,141].
14	GABRA5 (gamma-aminobutyric acid type A receptor subunit alpha5)	[142,143].
15	UBE3A (ubiquitin-protein ligase E3A)	[144,145,146].
16	CYFIP1 (cytoplasmic FMR1-interacting protein 1)	[147,148].
17	NRXN (neurexin)	[149,150].
18	STX1A (syntaxin 1A)	[151,152,153].
19	DLG4 (discs large MAGUK scaffold protein 4)	[154,155,156].
20	AKAP9 (A-kinase anchoring protein 9)	[157,158].
21	CACNA2D3 (calcium voltage-gated channel auxiliary subunit alpha2delta 3)	[159,160,161].
22	GSTT1(glutathione S-transferase theta-1), GSTM1(glutathione S-transferase Mu 1), GSTP1(glutathione S-transferase pi 1)	[162,163].
23	FMR1 (fragile X mental retardation 1)	[164,165,166].
24	APOE (apolipoprotein E)	[167,168,169,170].
25	APP (beta-amyloid precursor protein)	[171,172].
26	BDNF (brain-derived neurotrophic factor)	[173,174,175,176].
27	TNF (tumor necrosis factor)	[177,178].
28	SLC6A4 (solute carrier family 6 (neurotransmitter transporter, serotonin), member 4)	[179,180].

## Data Availability

Not applicable.

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
