# Peer review of "Symptomatic, Genetic, and Mechanistic Overlaps between Autism and Alzheimer’s Disease"

_biomolecules, 2021, doi:10.3390/biom11111635_

Round 1
Reviewer 1 Report
In this manuscript, the authors describe the results of a systematic review for the identification of common symptoms, genetics, and overlapping aspects in the mechanisms of pathogenesis of ASD and AD. It is a very interesting study summarizing the latest findings from different aspects.
There are few points that the authors should address to improve their manuscript:
- Correct the English grammar and syntax
- The authors are advised to use and submit as a supplementary file the PRISMA checklist for such a systematic review and specify the MeSH terms they used for they research and additional rephrase the following “All the data was collected from online data banks including Yahoo, Google Scholars, PubMed, and Web of Science. For data collection, terms like autism, Alzheimer’s disease, genes in autism, genes in Alzheimer’s disease, antidepressant, antipsychotic, sleep promoters, mechanism of autism pathogenesis, and mechanism of AD pathogenesis were used. A huge amount of recent data were collected from online search engines, data comprising of peer reviewed research articles published in the reputed journals was included in the further analysis”.
- Even though the literature is up-to-date the authors should clearly state the overlapping between Dementia, AD (sporadic and familial), ASD, and probably other neurodegenerative disorders, in lines 99-111.
The following references on AD and Dementia classification will be helpful:
- Dubois, B., Feldman, H. H., Jacova, C., Dekosky, S. T., Barberger-Gateau, P.,Cummings, J., et al. (2007). Research criteria for the diagnosis of Alzheimer’s disease: revising the NINCDS-ADRDA criteria. Lancet Neurol. 6, 734–746. doi: 10.1016/S1474-4422(07)70178-3
- Dubois, B., Feldman, H. H., Jacova, C., Hampel, H., Molinuevo, J. L., Blennow, K., DeKosky, S. T., Gauthier, S., Selkoe, D., Bateman, R., Cappa, S., Crutch, S., Engelborghs, S., Frisoni, G. B., Fox, N. C., Galasko, D., Habert, M. O., Jicha, G. A., Nordberg, A., Pasquier, F., … Cummings, J. L. (2014). Advancing research diagnostic criteria for Alzheimer's disease: the IWG-2 criteria. The Lancet. Neurology, 13(6), 614–629. https://doi.org/10.1016/S1474-4422(14)70090-0
- Alexiou, A., Mantzavinos, V. D., Greig, N. H., & Kamal, M. A. (2017). A Bayesian Model for the Prediction and Early Diagnosis of Alzheimer's Disease. Frontiers in aging neuroscience, 9, 77. https://doi.org/10.3389/fnagi.2017.00077
Author Response
Reviewer 1
Comments and Suggestions for Authors
In this manuscript, the authors describe the results of a systematic review for the identification of common symptoms, genetics, and overlapping aspects in the mechanisms of pathogenesis of ASD and AD. It is a very interesting study summarizing the latest findings from different aspects.
Author’s response:
The authors are grateful to the reviewer for sparing time and for valuable suggestions to improve the manuscript.
There are few points that the authors should address to improve their manuscript:
- Correct the English grammar and syntax
Author’s response:
The authors have a vast experience of article writing and they have tried their level best to improve the quality of English language in the manuscript. We have also applied online software (such as grammarly and ginger), to improve the English language.
- The authors are advised to use and submit as a supplementary file the PRISMA checklist for such a systematic review and specify the MeSH terms they used for they research and additional rephrase the following “All the data was collected from online data banks including Yahoo, Google Scholars, PubMed, and Web of Science. For data collection, terms like autism, Alzheimer’s disease, genes in autism, genes in Alzheimer’s disease, antidepressant, antipsychotic, sleep promoters, mechanism of autism pathogenesis, and mechanism of AD pathogenesis were used. A huge amount of recent data were collected from online search engines, data comprising of peer reviewed research articles published in the reputed journals was included in the further analysis”.
Author’s response
According to the recent international criteria for the diagnosis of AD, the disease do not need to interfere any occupational or social functioning at early stages. However, gradual impairments in memory and cognitive functions are prominent signs of AD [79]. Similar diagnostic and staging procedures and biomarkers have been used for AD and early memory impairment [80]. Structural disintegration and loss of neuronal connections have been recently identified by magnetic resonance imaging (MRI). Studies have also shown the association of physical activity to ameliorate the decline in the functioning of white matter and AD progression [81]. The paragraph above has been rephrased as directed by the reviewer. The changes/corrections have been incorporated according to the advice, PRISMA scheme has also been introduced in the methodology section.
|
||||||||||
PRISMA scheme
- Even though the literature is up-to-date the authors should clearly state the overlapping between Dementia, AD (sporadic and familial), ASD, and probably other neurodegenerative disorders, in lines 99-111.
The following references on AD and Dementia classification will be helpful:
- Dubois, B., Feldman, H. H., Jacova, C., Dekosky, S. T., Barberger-Gateau, P.,Cummings, J., et al. (2007). Research criteria for the diagnosis of Alzheimer’s disease: revising the NINCDS-ADRDA criteria. Lancet Neurol. 6, 734–746. doi: 10.1016/S1474-4422(07)70178-3
- Dubois, B., Feldman, H. H., Jacova, C., Hampel, H., Molinuevo, J. L., Blennow, K., DeKosky, S. T., Gauthier, S., Selkoe, D., Bateman, R., Cappa, S., Crutch, S., Engelborghs, S., Frisoni, G. B., Fox, N. C., Galasko, D., Habert, M. O., Jicha, G. A., Nordberg, A., Pasquier, F., … Cummings, J. L. (2014). Advancing research diagnostic criteria for Alzheimer's disease: the IWG-2 criteria. The Lancet. Neurology, 13(6), 614–629. https://doi.org/10.1016/S1474-4422(14)70090-0
- Alexiou, A., Mantzavinos, V. D., Greig, N. H., & Kamal, M. A. (2017). A Bayesian Model for the Prediction and Early Diagnosis of Alzheimer's Disease. Frontiers in aging neuroscience, 9, 77. https://doi.org/10.3389/fnagi.2017.00077
Author’s response
Many thanks for the suggestions. Some text related to the suggestions has been incorporated in the main manuscript,
According to the recent international criteria for the diagnosis of AD, the disease do not need to interfere any occupational or social functioning at early stages. However, gradual impairments in memory and cognitive functions are prominent signs of AD [79]. Similar diagnostic and staging procedures and biomarkers have been used for AD and early memory impairment [80]. Structural disintegration and loss of neuronal connections have been recently identified by magnetic resonance imaging (MRI). Studies have also shown the association of physical activity to ameliorate the decline in the functioning of white matter and AD progression [81].
All the above references have been incorporated as indicated in the red font as below:
- Dubois, B.; Feldman, H.H.; Jacova, C.; Dekosky, S.T.; Barberger-Gateau, P.; Cummings, J.; Research criteria for the diagnosis of Alzheimer’s disease: revising the NINCDS-ADRDA criteria. Lancet Neurol. 2007, 6, 734-746, https:// doi:10.1016/S1474-4422(07)70178-3.
- Dubois, B; Feldman, H.H.,; Jacova, C.; Hampel, H.; Molinuevo, J.L.; Blennow, K.; DeKosky, S.T.; Gauthier, S.; Selkoe, D.; Bateman, R.; Cappa, S.; Crutch, S.; Engelborghs, S.; Frisoni, G.B.; Fox, N.C.; Galasko, D.; Habert, M.O.; Jicha, G.A.; Nordberg, A.; Pasquier, F.; Cummings, J.L. Advancing research diagnostic criteria for Alzheimer's disease: the IWG-2 criteria. Neurol. 2014, 13, 614-629, https://doi.org/10.1016/S1474-4422(14)70090-0.
- Alexiou, A.; Mantzavinos, V.D.; Greig, N.H.; Kamal, M.A. A Bayesian Model for the Prediction and Early Diagnosis of Alzheimer's disease. Aging Neurosci. 2017, 9, 77, https://doi.org/10.3389/fnagi.2017.00077.

Reviewer 2 Report
I fundamentally disagree that Autism is a form of dementia. There is no progressive decline with aging. The authors should refer to current DSM-5 definitions and criteria.
Diagnostic Criteria for Dementia
DSM-5: Neurocognitive Disorder
Dementia is the umbrella term for a number of neurological conditions, of which the major symptom is the decline in brain function due to physical changes in the brain. It is distinct from mental illness.
New diagnostic criteria for dementia were developed and released in 2013.
Dementia is categorised as a Neurocognitive Disorder (NCD) in the Diagnostic and Statistical Manual of Mental Disorders (DSM-5). The NCD category is then further subdivided into Minor NCD and Major NCD. The term “cognitive” refers to thinking and related processes, and the term “neurocognitive” has been applied to these disorders to emphasise that brain disease and disrupted brain function lead to symptoms of NCD.
The NCD category encompasses the group of disorders that the primary clinical deficit is in cognitive function, which is acquired rather than developmental. Impairment may occur in attention, planning, inhibition, learning, memory, language, visual perception, spatial skills, social skills or other cognitive functions.
Author Response
Please find the attached response file

Reviewer 3 Report
The article is devoted to an actual problem of the correlated symptoms, genetics and molecular mechanisms in the pathogenesis of autism spectrum disorder and Alzheimer’s disease.
P. 1L. 3 Should be “Alzheimer’s disease”
P. 3 L. 105. Autism spectrum disorder (ASD) is a neurodevelopmental condition characterized by early onset of communication problems, including verbal communication, stereotypical behavior and restricted interests. Dementia is not typical for 100% of autistic patients. ASD - ID and ASD + ID are different subtypes of the disorder. Add the percentage of ASD - ID and ASD + ID (Intellectual Disability).
P. 3 L. 130 “Having poor cognitive skills is a prominent characteristic of children with autism.”
It is not true. Again, add the percentage of ASD + ID
P. 6 L. 223 “The Rett’s syndrome perfectly presenting this condition is associated with ASD”
ASD are commonly recognized as highly heterogeneous in their phenotypes, genetics, and etiology. Dysregulation of the mTORC1 pathway has been identified in numerous ASD syndromes, such as fragile X syndrome, tuberous sclerosis, PTEN hamartoma tumor syndrome, Rett syndrome, and Phelan–McDermid syndrome. The most part of both syndromic and idiopathic autism cases can be attributed to disorders caused by mTOR-dependent translation upregulation (Winden KD, Ebrahimi-Fakhari D, Sahin M. Abnormal mTOR Activation in Autism. Annu Rev Neurosci. 2018. doi: 10.1146/annurev-neuro-080317-061747; Onore C, Yang H, Van de Water J, Ashwood P. Dynamic Akt/mTOR Signaling in Children with Autism Spectrum Disorder. Front Pediatr. 2017. doi: 10.3389/fped.2017.00043).
Rett syndrome accounts for less than 1% of ASD, occurs almost exclusively in girls, while autism is diagnosed 4 times more often in boys; mTORC1 pathway is downregulated in Rett syndrome (Rangasamy S, Olfers S, Gerald B, Hilbert A, Svejda S, Narayanan V. Reduced neuronal size and mTOR pathway activity in the Mecp2 A140V Rett syndrome mouse model. F1000Res. 2016. doi: 10.12688/f1000research.8156.1). Altogether, it hardly contribute to the perception of Rett syndrome as a relevant model of the complete autistic spectrum. Please, revise the sentence.
P. 7 L. 232 “Imbalanced neurotransmitters”
Glutamate is the most important neurotransmitter presented in over 90% of all brain synapses. It should be mentioned in the context of ASD and Alzheimer’s disease pathophysiology (Khlebodarova TM, Kogai VV, Trifonova EA, Likhoshvai VA. Dynamic landscape of the local translation at activated synapses. Mol Psychiatry. 2018. doi: 10.1038/mp.2017.245; Eltokhi A, Santuy A, Merchan-Perez A, Sprengel R. Glutamatergic Dysfunction and Synaptic Ultrastructural Alterations in Schizophrenia and Autism Spectrum Disorder: Evidence from Human and Rodent Studies. Int J Mol Sci. 2020. doi: 10.3390/ijms22010059; Srivastava A, Das B, Yao AY, Yan R. Metabotropic Glutamate Receptors in Alzheimer's Disease Synaptic Dysfunction: Therapeutic Opportunities and Hope for the Future. J Alzheimers Dis. 2020. doi: 10.3233/JAD-201146)
P. 8 L. 286 Should be “export of”

Author Response
Please find the attached response file

Round 2
Reviewer 2 Report
OK